# Molecular Characteristics and Prevalence of Rifampin Resistance in *Staphylococcus aureus* Isolates from Patients with Bacteremia in South Korea

**DOI:** 10.3390/antibiotics12101511

**Published:** 2023-10-04

**Authors:** Yong Kyun Kim, Yewon Eom, Eunsil Kim, Euijin Chang, Seongman Bae, Jiwon Jung, Min Jae Kim, Yong Pil Chong, Sung-Han Kim, Sang-Ho Choi, Sang-Oh Lee, Yang Soo Kim

**Affiliations:** 1Department of Internal Medicine, Division of Infectious Diseases, Hallym University Sacred Heart Hospital, Hallym University College of Medicine, Anyang 14068, Republic of Korea; amoureuxyk@hallym.or.kr; 2Division of Infectious Diseases, Asan Medical Center, University of Ulsan College of Medicine, Seoul 05505, Republic of Korea; xenkins0618@gmail.com (E.C.); trueblue27@naver.com (J.J.); nahani99@gmail.com (M.J.K.); drchong@amc.seoul.kr (Y.P.C.); shkimmd@amc.seoul.kr (S.-H.K.); sangho@amc.seoul.kr (S.-H.C.); soleemd@amc.seoul.kr (S.-O.L.); 3Center for Antimicrobial Resistance and Microbial Genetics, University of Ulsan College of Medicine, Seoul 05505, Republic of Korea; 95tkrhk@naver.com (Y.E.); mykes@hanmail.net (E.K.); rarypooh@naver.com (S.B.); 4Asan Medical Center, Asan Institute for Life Science, Seoul 05505, Republic of Korea

**Keywords:** rifampin resistance, *Staphylococcus aureus*, bacteremia, molecular typing, *rpoB* gene, mutation

## Abstract

Rifampin resistance (RIF-R) in *Staphylococcus aureus* (*S. aureus*) with *rpoB* mutations as one of its resistance mechanisms has raised concern about clinical treatment and infection prevention strategies. Data on the prevalence and molecular epidemiology of RIF-R *S. aureus* blood isolates in South Korea are scarce. We used broth microdilution to investigate RIF-R prevalence and analyzed the *rpoB* gene mutation in 1615 *S. aureus* blood isolates (772 methicillin-susceptible and 843 methicillin-resistant *S. aureus* (MRSA)) from patients with bacteremia, between 2008 and 2017. RIF-R prevalence and antimicrobial susceptibility were determined. Multilocus sequence typing was used to characterize the isolate’s molecular epidemiology; Staphylococcus protein A (s*pa*)*,* staphylococcal cassette chromosome *mec* (SCC*mec*), and *rpoB* gene mutations were detected by PCR. Among 52 RIF-R MRSA isolates out of 57 RIF-R *S. aureus* blood isolates (57/1615, 0.4%; 5 methicillin-susceptible and 52 MRSA), ST5 (44/52, 84.6%), SCC*mec* IIb (40/52, 76.9%), and *spa* t2460 (27/52, 51.9%) were predominant. *rpoB* gene mutations with amino acid substitutions showed that A477D (17/48, 35.4%) frequently conferred high-level RIF resistance (MIC > 128 mg/L), followed by H481Y (4/48, 8.3%). RIF-R *S. aureus* blood isolates in South Korea have unique molecular characteristics and are closely associated with *rpoB* gene mutations. RIF-R surveillance through *S. aureus*–blood isolate epidemiology could enable effective therapeutic management.

## 1. Introduction

*Staphylococcus aureus* (*S. aureus*) is one of the most notorious and clinically significant pathogens in both community-acquired and nosocomial infections that causes various illnesses, which range from relatively minor local infections to life-threatening systemic infections [1]. Widespread antibiotic use has contributed to the emergence of multidrug-resistant *S. aureus* strains, particularly those that are resistant to methicillin, and this has become a global concern [2]. Rifampin is an antimicrobial agent with activity against multidrug-resistant *S. aureus*, including methicillin-resistant *S. aureus* (MRSA). Thus, rifampin is currently indicated in combination therapy for *S. aureus* bacteremia, prosthetic joint infections, and prosthetic valve endocarditis [3,4,5]. However, in MRSA, rifampin resistance (RIF-R) has emerged and is attributed to *rpoB* gene mutations [6]. RIF-R prevalence has rapidly increased to create a high RIF-R resistance rate [7,8]. The molecular characteristics of RIF-R and *rpoB* mutations in clinical MRSA [8,9] and those of *S. aureus* bloodstream isolates [10,11,12] have been previously evaluated. However, in South Korea, an accurate understanding of these characteristics is lacking, as there is limited information on the molecular characteristics of RIP-R and *rpoB* mutations in *S. aureus* infections. A recent study identified rifamycin resistance and *rpoB* gene mutations in *Staphylococcus* species isolates from prosthetic joint infections [13]; nonetheless, there is a paucity of data on the molecular epidemiology and prevalence of RIF-R and *rpoB* mutations in *S. aureus* isolates from patients with bacteremia in South Korea. With regard to treatment and infection control, the dissemination of RIF-R isolates in *S. aureus* bacteremia may pose a serious threat, and this highlights the importance of understanding the molecular epidemiology of *S. aureus* isolates.

Therefore, we aimed to investigate the prevalence of RIF-R and analyze the *rpoB* gene mutation in *S. aureus* isolates obtained from patients with bacteremia.

## 2. Results

### 2.1. Characteristics of Study Isolates and Profile of rpoB Gene Mutations 

The study sample comprised 1615 *S. aureus*, 843 MRSA, and 772 MSSA isolates obtained from patients with bacteremia. Among these, 57 (57/1615, 0.4%) were resistant to RIF. Antibiotic susceptibility testing using the broth microdilution method revealed that 52 RIF-R *S. aureus* isolates (52/57, 91.2%) were MRSA, and five RIF-R *S. aureus* isolates (5/57, 8.8%) were methicillin-susceptible *S. aureus* (MSSA). RIF resistance was more common in MRSA than in MSSA (*p* < 0.001) isolates.

Among the 52 RIF-R MRSA isolates, 51 isolates (51/52, 98.1%) showed high-level rifampin resistance (MIC ≥ 8 mg/L), and 1 isolate (1/52, 1.9%) showed low-level rifampin resistance (MIC 4 mg/L) (Figure 1). No mutations in the rifampin resistance-determining region (RRDR) of the *rpoB* gene were detected in 4 of the 52 RIP-R MRSA isolates (4/52, 7.7%), though these isolates were phenotypically resistant to RIF. Amplification of the 714-bp RRDR region of the *rpoB* gene processed for DNA sequence analysis and the MIC distributions for RIP related to *rpo*B mutations are shown in Table 1. *rpoB* gene mutations were detected in 41 out of 48 (41/48, 85.4%) isolates compared with the *rpoB* gene sequence of the reference strain. We identified 19 different types of mutations in *rpoB* gene mutation analysis, with 33 single mutations (33/48, 68.8%) and 15 multiple mutations (15/48, 31.3%). Of the 48 isolates, 17 (17/48, 35.4%) showed a mutation at codon 477 (gct to gat) and presented the amino acid substitution A477D, which was associated with high-level RIF resistance (MIC > 128 mg/L). We found several amino acid substitutions of H481Y and S468P that were common mutations of the rpoB proteins and some relatively rare mutations of R484H, S486L, and D471G that some previous studies have reported [2,8]. In addition, some isolates in our study carried rare mutations in RRDR, such as Q486L, Q486R, H481R, E568K, and R197L. However, when we sequenced 50 RIF-S MRSA strains as a negative control, we found that there was no detection of amino acid substitutions in any of the 50 RIF-S *S. aureus* isolates. 

### 2.2. Antimicrobial Resistance Profiles of RIF-R MRSA Isolates

The antibiotic resistance profiles of the 52 RIF-R MRSA isolates are shown in Figure 2. Our results revealed that the resistance levels of 52 RIF-R MRSA isolates were as follows: 100% to ampicillin, 94.2% to clindamycin, 92.3% to ciprofloxacin, 94.2% to erythromycin, 76.9% to fusidic acid, 73.1% to gentamicin, 100% to oxacillin, 100% to penicillin, 71.1% to tetracycline, 0% to quinupristin, and 7.7% to trimethoprim–sulfamethoxazole (TMP-SMX).

### 2.3. Molecular Typing

Among the total of 1615 *S. aureus* isolates (843 MRSA and 772 MSSA), ST5 was the predominant ST (543/1615, 33.6%), followed by ST72 (403/1615, 25.0%). Of the 57 RIF-R *S. aureus* isolates (52 MRSA and 5 MSSA), ST5 was predominant (44/57, 77.2%), and there were 6 non-ST5 isolates (1 ST72, 2 ST254, 1 ST1, 1 ST6, 1 ST101, and 1 ST45). In 1558 RIF-S *S. aureus* isolates (791 MRSA and 767 MSSA), 463 ST5 (29.7%) in MRSA and 137 ST72 (17.9%) in MSSA were the predominant STs.

Staphylococcal chromosomal cassette *mec* (SCC*mec*) typing, MLST, and spa typing were performed on the 52 RIF-R MRSA strains (Table 2). SCC*mec* typing revealed that the most common type was SCC*mec* type IIb (40/52, 76.9%), followed by IVa (7/52, 13.5%), Ⅰc (2/52, 3.8%), II (2/52, 3.8%), and IIa (1/52, 1.9%). The results of spa typing revealed a diverse distribution of the 52 RIF-R MRSA strains. The most predominant spa type was t2460 (27/52, 51.9%), followed by t002 (4/52, 7.7%) and t324 (4/52, 7.7%). Four spa-type isolates remained undetermined. 

We found that the predominant RIF-R MRSA was the ST5-Ⅱb-t2460 (26/52, 50%) molecular type, which had high resistance to RIF. Among the MSSA isolates, ST72-I-t126 (2/5, 40%) isolates from persistent carriers were resistant to RIF.

### 2.4. δ-Hemolysin Activities As Genotypic Characteristics of S. aureus Isolates by Rifampin

Table 3 shows the differences in the genotypic characteristics of *S. aureus* isolates with RIF susceptibility and resistance. The *agr* functionality test was performed on 1615 *S. aureus* isolates. Fourteen (24.6%) functional *agr* (hemolytic strains) were resistant to RIF, and 835 (53.6%) *agr* (hemolytic strains) were susceptible to RIF. There were 43 (75.4%) dysfunctional *agr* (nonhemolytic strains) resistant to RIF and 723 (46.4%) dysfunctional *agr* (nonhemolytic strains) susceptible to RFP.

## 3. Discussion

In this study, we evaluated the molecular characteristics and prevalence of RIF resistance in *S. aureus* isolates obtained from patients with bacteremia. Our results suggested that RIF resistance was detected in 0.4% (57/1615) of *S. aureus* isolates associated with bloodstream infections, and ST5-SCC*mec* Ⅱb-*spa*t 2460 was the most common type of RIF-R *S. aureus* isolate. Amino acid A477D substitution was most common in RIF-R MRSA isolates that harbored multiple mutations, which conferred high-level RIF resistance (MIC > 128 mg/L). Notably, the RIF-R MRSA isolates showed high rates of resistance to various antibiotics.

Several recent studies have reported the geographically diverse molecular epidemiology of prevalent MRSA clones in bacteremia, with a high prevalence of ST5 and ST59 (People’s Republic of China) [14,15,16], ST 239 (Turkey) [12], ST239 (Iran and Kuwait) [11,17], and ST1011 (Mexico) [10]. However, limited information is available regarding the molecular epidemiology of *S. aureus* in patients with bacteremia in South Korea. ST5-SCC*mec*II and ST239-SCC*mec*III clones were predominant among a relatively small number of blood isolates (*n*, 45–96) [18,19]. Community-associated (CA) MRSA strains of ST72-SCC*mec* IV accounted for a major proportion of MRSA blood isolates (from 76 to 83) in South Korea [20,21,22]. A key strength of our study was the evaluation of a large number of *S. aureus* blood isolates (*n* = 1615); ST5-SCC*mec*II (543/1615, 33.6%) and ST72-SCC*mec*IV (403/1615, 25.0%) were the major clones in South Korea, which reinforced the results of previous molecular epidemiology studies. To the best of our knowledge, this is the first study to evaluate the molecular characteristics of RIF-R *S. aureus* blood isolates in South Korea, and we identified molecular characteristics distinct from those reported in previous studies on RIF-R *S. aureus* clinical isolates [8,9,13,23].

Mutations conferring RIF-R are almost exclusively confined to the *rpoB* gene in most microorganisms [24], and *rpoB* mutations can be associated with high-level RIF resistance [25]. In our study, we amplified and sequenced portions of *rpoB* from RIF-R *S. aureus* isolates. High-level RIF resistance may be attributed to additional mutations within the *rpoB* gene, as previously described [26]. Though sequencing of the RRDR successfully identified mutations in RIF-R MRSA isolates, four RIF-R MRSA isolates (4/52, 7.7%) lacked any mutation in the RRDR region of the *rpoB* gene despite phenotypic resistance to RIF. This difference might persist because of genotypic variations that are prevailing worldwide, and the presence of mutations in the *rpoB* gene other than in the RRDR region could not be excluded [27]. The lack of RRDR mutations in RIF-R MRSA isolates may be due to the presence of other rare *rpoB* mutations or another mechanism of RIF-R [27], which is supported by previous reports from several countries [28,29,30]. We posit that the four isolates (7.7%) likely possessed additional mutations outside the RRDR because they exhibited higher RIF-R levels than those conferred by the mutations in their RRDR alleles. On the contrary, 15 multiple mutations (15/48, 31.3%) were identified in *rpoB* gene mutation analysis in our study, all of which were associated with high-level RIF-R (MIC ≥ 8 mg/L). Although most *rpoB* gene mutations were correlated with high-level resistance [6], it is difficult to identify how much each mutation contributed to the increased MIC. Further molecular analysis of *rpoB* gene mutations in RIF-R *S. aureus* isolates to identify the correlation between high-level MIC and mutation at specific codons will provide higher robustness on isolates’ ability to survive and affect clinical practice in the region of endemicity. 

The amino acid substitution A477D was the most prevalent in our study, which was inconsistent with findings reported from previous studies that showed the H481Y MRSA clone was predominant [8,9,13,31]. Furthermore, the transformation of A477D-mutated *rpoB* into wild-type *S. aureus* strains resulted in a RIF-R phenotype, indicating that these mutations contribute to rifampicin resistance in *S. aureus*. In the A477D mutant, this substitution places a negatively charged carboxylate unit in close proximity to an existing carboxylate from H481, which is situated near the protein–DNA interface and thereby increases the negative charge on the surface of the protein and destabilizes the enzyme–DNA interaction, owing to electrostatic repulsion [32]. Even if the abovementioned substitution does not face the RIF target region, it can induce a conformational change that indirectly prevents antibiotic binding to the target site, which determines the RIF-R [8]. Another interesting finding of our study was that the analyzed isolates lacked the L466S mutation, which was one of the most commonly found mutations in the clinical isolates of other studies [7,9]. We posit that there should be regional differences in the mutation codon of RIF-R blood *S. aureus* isolates, given that only limited information is available regarding the molecular characteristics of *rpoB* mutations of blood *S. aureus* isolates in multiple countries, including South Korea. Future multi-center studies using isolates of national or global pathogen resource networks to further characterize the mutations associated with RIF-F and investigate new mutations are essential. 

*rpoB* mutations are associated not only with resistance to RIF but also resistance to other last-line antibiotics for MRSA, including vancomycin and daptomycin [33,34]. In particular, the reduced susceptibility to vancomycin that is linked to *rpoB* mutations may result in poor clinical outcomes and persistent MRSA infections [35]. We posit that the potentially decreased susceptibility to vancomycin promoted persistent infection which was associated with the *rpoB*-A477D mutation; this aspect may be a critical concern in the treatment of MRSA bacteremia because the A477D mutant was most prevalent in our 52 RIF-R MRSA isolates from patients with bacteremia [2,33,34]. In addition, given that RIF exposure can be the main selective pressure for decreased susceptibility to RIF and vancomycin and that a higher prevalence of RIF resistance in ST5 was prevalent in previous large-scale *S. aureus* genome studies [2], judicious use of RIF and preventive infection-control measures against the spread of RIF-R *S. aureus* clones should be prioritized based on large-scale genomic surveillance in the fight against *S. aureus* bacteremia. Further research is required to understand whether cross-resistance to other antibiotics develops during prolonged RIF exposure or whether resistance rapidly evolves during exposure to antibiotics [36].

Our study has certain limitations. First, this study was conducted at a single center. Second, we analyzed the clinical outcomes and risk factors of RIF-R *S. aureus* isolates obtained from patients with bacteremia. Third, we did not evaluate the molecular characteristics of the RIF-R MSSA isolates. We posit that our study could be strengthened further if we had sequenced more sufficient RIF-R and RIF-S *S. aureus* isolates, and it has limitations in being able to acquire strong confirmation that RIF-R in *S. aureus* isolates was caused by a *rpoB* mutation. However, our study included a relatively large number of *S. aureus* isolates from bacteremia (*n* = 1615) with RIF-R MRSA isolates (*n* = 52) and was conducted in a 2700-bed tertiary-care teaching hospital where patients from across the country were admitted during a 10-year period. Therefore, our study has important clinical implications for antibiotic use and provides useful information for preventive infection-control measures based on the molecular epidemiology of RIF-R *S. aureus* blood isolates in South Korea. 

## 4. Materials and Methods

### 4.1. Collection of S. aureus Isolates

The study sample consisted of *S. aureus* strain isolates obtained from Asan Medical Center, Seoul, Republic of Korea, from 2008 to 2017. The *S. aureus* samples were plated on a blood agar plate. This sterile medium was streaked with a cotton swab, and the plates were incubated overnight at 37 °C. The isolate was grown to screen for and analyze *S. aureus.* The strains were stored in 20% glycerol-tryptic soy broth at −80 °C (Becton Dickinson, Sparks, MD, USA). We determined the methicillin resistance of *S. aureus* isolates based on the oxacillin MIC and the presence of the *mecA* gene. 

### 4.2. Agr Functionality Test

We used δ-hemolysin activity to determine *agr* functionality by cross-streaking vertically to RN4220 and a test strain on a sheep blood agar plate (BAP). The β-hemolysin produced by RN4220 enables detection of δ-hemolysin [37]. δ-hemolytic activity was indicated by an enhanced area of hemolysis at the intersection of the streaks. 

### 4.3. Antimicrobial Susceptibility Tests

The antimicrobial susceptibility profiles of the *S. aureus* isolates were determined using the broth microdilution method according to the Clinical and Laboratory Standard Institute (CLSI) guidelines [38]. For the broth microdilution method, serial two-fold dilutions were carried out in cation-adjusted Mueller–Hinton II broth (Becton Dickinson, Sparks, MD, USA) in microtiter plates according to standard criteria [38]. The MIC was determined using the broth microdilution method, and each spot was inoculated with 10^6^ CFU. After incubation at 37 °C for 16–20 h, the MIC value was considered when the bacteria did not grow at the minimum antibiotic concentration. The reference strain from the American Type Culture Collection 29,213 was used for quality control. The MIC for rifampin (Sigma, Darmstadt, Germany) was determined using the broth microdilution method, following standard criteria [38]. Based on the CLSI guidelines, the isolates were classified as RIF susceptible (MIC ≤ 1 mg/L) or RIF resistant (MIC ≤ 4 mg/L).

### 4.4. Molecular Typing

#### 4.4.1. Detection of the mecA Gene

The *mecA* gene sequence (532 bp) of all the MRSA isolates was amplified by PCR. The amplification of the *mecA* gene was achieved using *mecA1* (5′ AAA ATC GAT GGT AAA GGT TGG C 3′) and *mecA2* (3′ AGT TCT GCA GTA CCG GAT TTG C 5′) primers and sequence analysis. The PCR conditions involved an initial denaturation for 3 min, followed by 30 cycles at 94 °C for 30 s, 55 °C for 30 s, and 72 °C 30 s and a final extension at 72 °C for 4 min. The PCR products (10 μL) were separated on a 1% agarose gel in 0.5 × Tris-borate-EDTA buffer at 100 V and visualized using RedSafe.

#### 4.4.2. Multilocus Sequence Typing

Multilocus sequence typing (MLST) of the isolates was conducted by amplifying internal fragments of seven housekeeping genes of *S. aureus*, as described previously [39]. The fragments were amplified using the following primers (reference): carbamate kinase (arcC), shikimate dehydrogenase (aroE), glycerol kinase (glpF), guanylate kinase (gmk), phosphate acetyltransferase (pta), triosephosphate isomerase (tpi), and acetyl coenzyme A acetyltransferase (yqiL). Following the purification and sequencing of these genes, allele quantification and sequence typing were performed using a well-characterized online database (https://pubmlst.org/, accessed on 10 August 2023).

#### 4.4.3. SCCmec Typing

The SCC*mec* typing of the MRSA isolates was performed using the multiplex PCR method described by Oliveira and de Lencastre [40]. The eight loci (A through H) and specific pairs of primers for SCC*mec* types and subtypes I, II, III, and IV have been described previously [41]. The multiplex PCR conditions included an initial denaturation at 4 min, followed by 30 cycles at 94 °C for 30 s, 53 °C for 30 s, and 72 °C for 1 min as well as a final extension at 72 °C for 4 min. The PCR products (10 μL) were separated on a 1.8% agarose gel in 0.5 × Tris-borate-EDTA buffer at 135 V and visualized using RedSafe.

#### 4.4.4. Spa Typing

The Staphylococcus protein A (spa) variable repeat region from each MRSA isolate was amplified using simplex PCR oligonucleotide primers, as described previously [40,42]. The purified spa PCR products were sequenced, and the typing of *spa* was performed using the public *spa* database website (http://spa.ridom.de/, accessed on 10 August 2023) for all *S. aureus* isolates. 

### 4.5. PCR Detection of Rifampin Resistance-Associated Mutations

Template DNA for PCR was obtained using a WizPrep gDNA Mini Kit (Seongnam, Republic of Korea). Total DNA from *S. aureus* was purified and used as a template for PCR amplification. The rifampin resistance-determining region (RRDR) of the *rpoB* gene sequence, measuring 690 bp (nucleotides 1307–2020), was amplified using PCR. Amplification of *rpoB* was performed using RRDR1 (5′ TTC AAG ATA CTG AGT CTA TCA CAC C 3′) and *rpoB* RRDR2 (3′ GCA CG T GAT TCT GGT GCA GCT ATT A 5′) primers, followed by sequence analysis. Amplification was performed for 52 RIF-R and 50 RIF-S strains as previously described [43], and the 50 RIF-S strains were sequenced as a negative control: 5 oligonucleotide primers rpoB1-F (5′ ATG GTA TTT AGC TAA AAG CGG 3′), rpoB1-R (3′ GCA CTG AAA ACA CTG AAC AA 5′), rpoB2-F (5′ ATT AGG TTT CTC AAG TGA CC 3′), rpoB2-R (3′ CCA TTA GCT GAG TTA ACG CAT 5′), rpoB3-F (5′ AAG CAG TGC CTT TGA ATC C 3′), rpoB3-R (3′ CCT AAA GGT GTA ACT GAG TT 5′), rpoB4-F (5′ TTG GTG CAG AAG TAA AAG ATG G 3′), rpoB4-F (3′ GGT GTA ATG TAC ATG TTG AA 5′), rpoB5-F (5′ AAT CTT GGT ATT CAC GTT GC 3′), and rpoB5-R (3′ GCT GAA TTT TAT TGA TT 5′) were identified using for mutation. The PCR products were purified and analyzed using DNA sequencing. PCR was performed in a DNA thermal cycler (Thermo Fisher Scientific, Waltham, MA, USA). The *rpoB* PCR cycling programs consisted of an initial denaturation (4 min at 94 °C) followed by 35 cycles of denaturation (30 s at 94 °C), annealing (45 s at 53 °C), and extension (45 s at 72 °C), with a final extension (3 min at 72 °C). The except RRDR PCR cycling programs consisted of an initial denaturation (4 min at 94 °C) followed by 35 cycles of denaturation (45 s at 94 °C), annealing (45 s at 50 °C), and extension (1 min at 72 °C), with a final extension (10 min at 72 °C). DNA sequencing was performed by COSMO Genetech (Seoul, Republic of Korea). The nucleotide sequences obtained were compared to the *rpoB* wild-type sequence from *S. aureus* subsp. aureus (GenBank accession number: N315) using clustalw software (https://www.genome.jp/tools-bin/clustalw, accessed on 10 August 2023).

### 4.6. Statistical Analysis

The statistical distribution and clinical characteristics of the patients were compared using cross-analysis and the chi-square test, respectively. Statistical significance was set at *p* < 0.05. All statistical analyses were performed using SPSS version 24.0 (SPSS Inc., Chicago, IL, USA). 

## 5. Conclusions

We showed that 0.4% of *S. aureus* blood isolates (57/1615) demonstrated RIF resistance in a tertiary care hospital in South Korea during a 10-year period. The RIF-R *S. aureus* blood isolates had unique molecular characteristics, revealing that RIF-R in *S. aureus* was highly related to mutations in *rpoB* and ST5-SCC*mec* Ⅱb-*spa*t 2460, with high-level RIF resistance being the most predominant type. A surveillance system based on the molecular epidemiology of *S. aureus* blood isolates should be implemented in hospitals to improve clinical treatment and infection prevention strategies. 

## Figures and Tables

**Figure 1 antibiotics-12-01511-f001:**
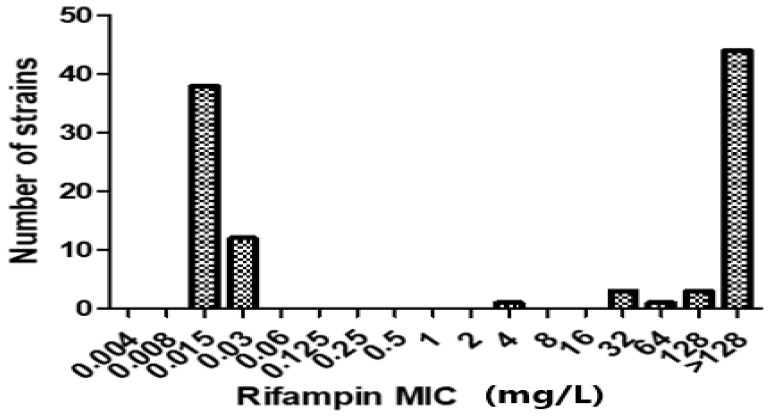
Distribution of rifampin (RIF) minimum inhibitory concentrations (MICs) for 52 RIF-R MRSA isolates. Of the 52 isolates, 51 (98.1%) showed high-level RIF resistance (MIC ≥ 8 mg/L), and only 1 (1.9%) had low-level RIF resistance (MIC 4 mg/L). Additionally, the broth microdilution method was used to determine the MIC distributions of 50 RIF-susceptible MRSA isolates.

**Figure 2 antibiotics-12-01511-f002:**
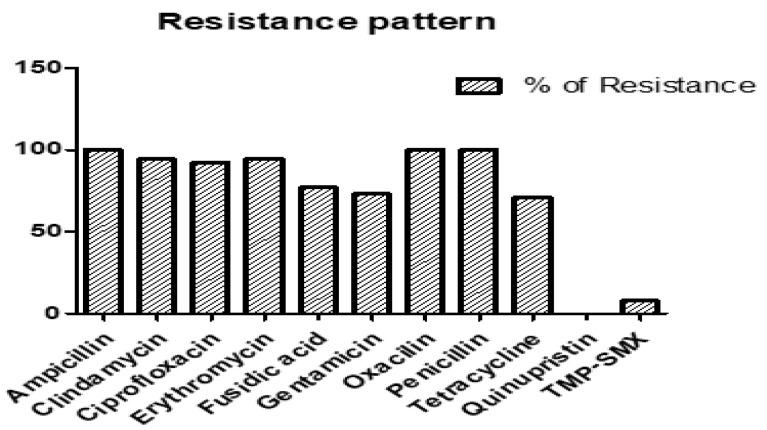
Percentage of antibiotic resistance in 52 rifampin (RIF)-R MRSA isolates.

**Table 1 antibiotics-12-01511-t001:** Correlation of *rpoB* gene mutations and the level of resistance to rifampin in MRSA isolates.

MRSA *rpoB* Mutations		Rifampicin MIC	
Nucleotide Mutation	Amino Acid Substitution	MIC μg/mL	Number of Isolates
gct/gat	A477D	>128	17
cat/tat	H481Y	>128	4
agc/aac + tct/cct	S463N, S464P	>128	2
caa/cta	Q468L	>128	2
gct/gat + act/gct	A477D, T801A	>128	2
cgt/cat	R484H	>128	1
tct/cct + att/ctt	S464P, I527L	>128	1
caa/cga	Q468R	>128	1
cat/aat + att/atg	H481N, I527M	>128	1
cat/aat + att/atg + gaa/aaa	H481N, I527M, E568K	>128	1
cat/aat + tca/tta	H481N, S529L	>128	1
cat/cgt	H481R	>128	1
gct/act + cat/aat	A477T, H481N	>128	1
tca/tta	S486L	>128	1
gac/ggc	D471G	>128	1
gac/gaa + gct/gat	D471E, A477D	>128	1
caa/cta + gat/gaa	Q468L, D668E	>128	1
caa/cta + aaa/ata	Q468L, K1166I	>128	1
att/ctt + gct/gat	I448L, A477D	>128	1
gaa/aaa	E568K	>128	1
agc/aac + tct/cct + aaa/ata	S463N, S464P, K1584I	128	1
cgt/ctt	R197L	128	1
cta/ata + acg/aag	L485I, T480K	64	1
tct/cct	S464P	32	2
att/cat	I527H	32	1

**Table 2 antibiotics-12-01511-t002:** Molecular characteristics of the 52 rifampin-resistant MRSA isolates.

MLST	SCCmec					Spa Type	Number of Isolates
	Ic	II	IIa	IIb	IVa		
ST5			1	25		t2460	26
		2		2		t002	4
				3		t9353	3
				2		t324	2
				1		t1228	1
				1		t2461	1
				1		t9363	1
				1		t264	1
				1		t564	1
				1		t18239	1
				2	1	unknown	3
ST72					1	t324	1
					1	t2431	1
					1	t664	1
					1	t148	1
					1	unknown	1
ST254	1					t324	1
	1					t688	1
ST1					1	t2460	1

**Table 3 antibiotics-12-01511-t003:** Genotypic characteristics of the *S. aureus* isolates stratified by their rifampin-resistance status.

Genotype	Number (%) of Isolates	
	Rifampin Resistance (*n* = 57)	Rifampin Susceptible (*n* = 1558)
*n* = 1615		
MRSA	52 (91.2) *	791 (48.9)
MSSA	5 (8.8) *	767 (47.5)
*n* = 1615		
agr function	14 (24.6) *	835 (53.6)
agr dysfunction	43 (75.4) *	723 (46.4)
*n* =1615		
ST5	44 (77.2) *	499 (32.0)
Non-ST5	13 (22.8) *	1059 (68.0)

* *p* < 0.001.

## Data Availability

The datasets generated and/or analyzed during the current study are available from the corresponding author upon reasonable request.

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
