# Peer review of "Molecular Characteristics and Prevalence of Rifampin Resistance in Staphylococcus aureus Isolates from Patients with Bacteremia in South Korea"

_antibiotics, 2023, doi:10.3390/antibiotics12101511_

Round 1

Reviewer 1 Report

Line 61 chapter 3 is coming after chapter 1, is it a typo? 

Introduction is 1, results is 3, discussion is 4 and the materials again 4.

Line 261 10uL? micro L, I presume

Introduction is short but sufficient

Results in a great way are not following the materials. There is no clear how your strains are divided in MRSA and MSSA, on the basic of what test?

Agr functionality is performed on all isolates? And the genetic test only on MRSA which is RIF-R? Why there are no controls? Groups that were MRSA and not RIF-R, MSSA RIF-R and non RIF-R were not tested on mutations? 

In this way you can not be secure that your S. aureus strains that are not RIF-R do not have similar mutations. You write that RIF-R possibly from rpoB mutation - line 18. You do not have the control group in this way that can be used as a confimation.

English is fine.

Author Response

Reviewer #1:

Comment 1:

Line 61 chapter 3 is coming after chapter 1, is it a typo? 

Introduction is 1, results is 3, discussion is 4 and the materials again 4.

Response 1: Thank you for this comment. When we made the structure of paragraphs and headings, there was mistake that should not be happened. Following the reviewer’s comment, we have revised the headings and sections in a correct order. 

Comment 2:

Line 261 10uL? micro L, I presume.         

Response 2: Thank you for this comment. The staphylococcal cassette chromosome mec (SCCmec) type was identified using previously described methods in our study [Oliveira DC et al. Antimicrob Agents Chemother 2002;46:2155-2161, Shopsin B et al. J Clin Microbiol 2003;41:456-459]. Following the reviewer’s comment, we have revised the manuscript as micro L. 

Comment 3:

Introduction is short but sufficient

Response 3: Thank you for this comment. We tried to set the context for the study, clearly state the research objective, and establish the significance of the study in Introduction.  

Comment 4:

Results in a great way are not following the materials. There is no clear how your strains are divided in MRSA and MSSA, on the basic of what test?

Response 4: Thank you for this valuable comment and we agree with the reviewer’s concern. In our study, methicillin resistance was determined based on MIC of oxacillin and the presence of mecA.

As the reviewer’s comment, we included the test that we used to determine the methicillin resistance in S. aureus isolates and revised the manuscript as follow.

1) To) added in Materials and Methods (4.1. Collection of S. aureus Isolates), P.7

The study sample consisted S. aureus strains isolates obtained from Asan Medical Center, Seoul, Republic of Korea, from 2008 to 2017. S. aureus samples were plated on a blood agar plate. This sterile medium was streaked with a cotton swab and the plates were incubated overnight at 37℃. The isolate was grown to screen for and analyze S. aureus. The strains were stored in 20% glycerol-tryptic soy broth at −80℃ (Becton Dickinson, Sparks, MD). We determined the methicillin resistance of S. aureus isolates based on the oxacillin MIC and the presence of the mecA gene.

Comment 5: Agr functionality is performed on all isolates? And the genetic test only on MRSA which is RIF-R? Why there are no controls? Groups that were MRSA and not RIF-R, MSSA RIF-R and non RIF-R were not tested on mutations? 

In this way you can not be secure that your S. aureus strains that are not RIF-R do not have similar mutations. You write that RIF-R possibly from rpoB mutation - line 18. You do not have the control group in this way that can be used as a confirmation.

Response 5: Thank you for this valuable comment and we totally agree with the reviewer’s concern.

Although we performed molecular typing of 1615 S. aureus isolates that revealed the predominance of ST5 (543/1615, 33.6%) and agr functionality on 1615 S. aureus isolates, the amplification of RRDR of the rpoB gene was performed only for 102 S. aureus isolates (52 RIF-R and 50 RIF-S MRSA strains) in our study. When we sequenced the 50 RIF-S MRSA stains as a negative control, we found that there was no detection of amino acid substitutions in any of the 50 RIF-S S. aureus isolates.

However, as the reviewer’s comment, we did not sequence 5 RIF-R MSSA and any of RIF-S MSSA isolates. We posit that our study could be more strengthened if we had sequenced sufficient RIF-R and RIF-S S. aureus isolates, and it has limitation to make a robust confirmation that RIF-R in S. aureus isolates was caused by rpoB mutation.

Following the reviewer’s comment, we tried to lend a more euphemistic tone to the text regarding the association between RIF-R and rpoB mutation, and have revised the manuscript as follow.

1) From) Abstract, P.1

Abstract: Rifampin resistance (RIF-R) in Staphylococcus aureus (S. aureus), possibly from rpoB mutations, has raised concern about clinical-treatment and infection-prevention strategies. Data on the prevalence and molecular epidemiology of RIF-R S. aureus blood isolates in South Korea are scarce.

To) Abstract, P.1

Abstract: Rifampin resistance (RIF-R) in Staphylococcus aureus (S. aureus) with rpoB mutations as one of its resistance mechanisms has raised concern about clinical-treatment and infection-prevention strategies. Data on the prevalence and molecular epidemiology of RIF-R S. aureus blood isolates in South Korea are scarce.

2) From) Results (2.1. Characteristics of Study Isolates and Profile of rpoB Gene Mutations), P.2

Of the 48 isolates, 17 (17/48, 35.4%) showed a mutation at codon 477 (gct to gat) and presented the amino acid substitution A477D, which was associated with high-level RIF resistance (MIC >128 mg/L). However, amino acid substitutions were not detected in any of the 50 RIF-S S. aureus isolates.

To) Results (2.1 Characteristics of Study Isolates and Profile of rpoB Gene Mutations), P.2

Of the 48 isolates, 17 (17/48, 35.4%) showed a mutation at codon 477 (gct to gat) and presented the amino acid substitution A477D, which was associated with high-level RIF resistance (MIC >128 mg/L). However, when we sequenced 50 RIF-S MRSA stains as a negative control, we found that there was no detection of amino acid substitutions in any of the 50 RIF-S S. aureus isolates.

3) To) added in Discussion, P.7

Our study has certain limitations. First, this study was conducted at a single center. Second, we analyzed the clinical outcomes and risk factors of RIF-R S. aureus isolates obtained from patients with bacteremia. Third, we did not evaluate the molecular characteristics of the RIF-R MSSA isolates. We posit that our study could be strengthened further if we had sequenced more sufficient RIF-R and RIF-S S. aureus isolates, and it has limitation in being able to acquire strong confirmation that RIF-R in S. aureus isolates was caused by a rpoB mutation. However, our study included a relatively large number of S. aureus isolates from bacteremia (n = 1615) with RIF-R MRSA isolates (n = 52) and was conducted in a 2700-bed tertiary-care teaching hospital where patients from across the country were admitted during a 10-year period.

4) To) added in Materials and Methods (4.5. PCR Detection of Rifampin Resistance-associated Mutation), P.8

The rifampin resistance-determining region (RRDR) of the rpoB gene sequence, measuring 690 bp (nucleotides 1307–2020), was amplified using PCR. Amplification of rpoB was performed using RRDR1 (5′ TTC AAG ATA CTG AGT CTA TCA CAC C 3′) and rpoB RRDR2 (3′ GCA CG T GAT TCT GGT GCA GCT ATT A 5′) primers, followed by sequence analysis. Amplification was performed for 52 RIF-R and 50 RIF-S strains as previously described [21], and the 50 RIF-S strains were sequenced as a negative control:

Thank you for your kind and helpful comments.

Reviewer 2 Report

This is a very nicely presented large work with a great scientific merit which has also practical implications.

On my opinion itcan be accepted as it is.

Congratulations!

Author Response

Reviewer #2:

This is a very nicely presented large work with a great scientific merit which has also practical implications.

On my opinion it can be accepted as it is.

Congratulations!

Response 1: Thank you for this comment. We tried to make a contribution to the literature identifying the unique molecular characteristics of RIF-R S. aureus blood isolates in South Korea and the potential role of the RIF-R of S. aureus in rpoB mutations.

For the future study, we believe that we have to conduct a multi-center study using isolates of a national a national pathogen resource network to further characterize the mutations associated with RIF-R and investigate new mutations as well as analysis of clinical outcomes and risk factors of RIF-R S. aureus isolates obtained from patients with bacteremia.

Thank you for your kind and helpful comments.

Reviewer 3 Report

The manuscript investigated the prevalence of rifampin resistant Staphylococcus Aureus in South Korea and pinpointed the correlation of rpoB mutations and the level of resistance to rifampin. The study revealed that rifampin resistant isolates also present resistance to other last line antibiotics, which provided critical reference for treating bacteremia. The writing is clear and the data is well presented. 

Could authors elaborate on the reason or hypotheses why rifampin resistant strains present resistance to other antibiotics, given these antibiotics (e.g. rifampin and vancomycin) utilize different mechanisms of action for their potency?

Author Response

Reviewer #3:

The manuscript investigated the prevalence of rifampin resistant Staphylococcus Aureus in South Korea and pinpointed the correlation of rpoB mutations and the level of resistance to rifampin. The study revealed that rifampin resistant isolates also present resistance to other last line antibiotics, which provided critical reference for treating bacteremia. The writing is clear and the data is well presented. 

Could authors elaborate on the reason or hypotheses why rifampin resistant strains present resistance to other antibiotics, given these antibiotics (e.g. rifampin and vancomycin) utilize different mechanisms of action for their potency

Response 1: Thank you for this valuable comment.

There are a few possible explanations why rifampin-resistant strains of S. aureus may also be resistant to other antibiotics. One reason is that rifampin resistance is often caused by mutations in the rpoB gene, which encodes the beta subunit of RNA polymerase. Because rpoB gene is essential for bacterial survival, mutations in this gene can lead to cross-resistance to other antibiotics. A number of studies have shown a relationship between rpoB mutations and decreased susceptibility to rifampin as well as to other last-line anti-MRSA drugs as vancomycin, daptomycin, beta-lactams, or imipenem [Matsuo M et al. Antimicrob Agents Chemother 2011;55:4188-4195, Watanabe Y et al. J Clin Microbiol 2011;49:2680-2684, Cui L et al Antimicrob Agents Chemother 2010;54:5222-5233, Aiba Y et al Antimicrob Agents Chemother 2013;57:4861-4871]. Another suggestion is that rifampin is often used in combination with other antibiotics to treat S. aureus infections, particularly those associated with biofilm formation. Rifampin-resistant strains of S. aureus may be more likely to form biofilms. Biofilms can be difficult to treat with antibiotics, and they may also promote the development of antibiotic resistance [Tang HJ et al. Antimicrob Agents Chemother 2013;57:5717-5720, Savage VJ et al. Antimicrob Agents Chemother 2013;57:1968-1970].

Thank you for your kind and helpful comments.

Reviewer 4 Report

Thank you for the opportunity to review the manuscript entitled “Molecular Characteristics and Prevalence of Rifampin Resistance of Staphylococcus aureus Isolates from Patients With Bacteremia in South Korea.” The study described the prevalence of rifampicin resistance among S. aureus and the association with different typing schemes. Overall the study was sound and interesting. Although there are a few points that may need to be clarified.

1.      It is stated that rpoB mutation is the common mechanism of rifampicin resistance in S. aureus. Can the authors further summarize the known (or common) mutations associated with the resistance? In this study, did the authors find any new and uncharacterized mutations?

2.      The authors only sequenced the RIF-R strains so there was no negative control in the study. Perhaps if the authors can sequence a few RIF-S strains to show that they do not have similar rpoB mutations, the study could be greatly strengthened.

3.      From points 1 and 2, in Table 1 there were a few strains with more than 1 mutation. It is rather difficult to identify which mutation (one or more) contributed to the increased MIC.

4.      Line 109 – 110: there are typos (followed by “a” [should be “IVa”], as well as another “a” below).

Some parts of the manuscript are very difficult to understand. For example, Lines 26 - 30 (abstract). Please consult language editors. Also, I noticed the use of the word "pluripotent pathogen", which is rather unusual. Can the authors confirm that this is the right word?

Author Response

Reviewer #4:

Thank you for the opportunity to review the manuscript entitled “Molecular Characteristics and Prevalence of Rifampin Resistance of Staphylococcus aureus Isolates from Patients With Bacteremia in South Korea.” The study described the prevalence of rifampicin resistance among S. aureus and the association with different typing schemes. Overall the study was sound and interesting. Although there are a few points that may need to be clarified.

Comment 1: It is stated that rpoB mutation is the common mechanism of rifampicin resistance in S. aureus. Can the authors further summarize the known (or common) mutations associated with the resistance? In this study, did the authors find any new and uncharacterized mutations?

Response 1: Thank you for this valuable comment.

As the reviewer’s comment, the common mechanism of the rifampin resistance in S. aureus is mutations in the rifampin resistance-determining region (RRDR) [Aubry-Damon H et al. Antimicrob Agents Chemother 1998;42:2590-2594]. Previous studies have revealed that there are common mutations that cause amino acid sequence changes such as V453F, S464P, L466S, D471N, A473D, A477D or T, H481N or Y, and I517L or M [Sekiguchi JI et al. J infect Chemother 2006;12:47-50, Wang B et al. Emerg Microbes Infect 2022;11:532-542].

In our study, amino acid substitutions of A477D, H481Y, and S464P were common mutations in the rpoB proteins, which was consistent with other studies. In addition, we found some relatively rare mutations of R484H, S486L, D471G that some other studies have also shown [Guérillot R et al. mSphere 2018;3:e00550-17, Bongiorno D et al. Microb Drug Resist 2018;24:732-738].

However, some isolates in our study carried mutations in RRDR, such as Q486L, Q486R, H481R, E568K, R197L, which few previous studies have reported. Another interesting finding of our study was that the analyzed isolates lacked the L466S mutation that were one of the most commonly found mutations in clinical isolates of other studies [Zhou W et al. BMC Microbiol 2012;12:240, Guo Y et al. Infect Drug Resist 2021;14:4591-4600].

We posit that there should be regional differences in the mutation codon of RIF-R blood S. aureus isolates, given that only limited information is available regarding the molecular characteristics of RIF-R and rpoB mutations of blood S. aureus isolates in multiple countries including South Korea. We are planning the future multi-center study using isolates of National Culture Collection for pathogens, a national pathogen resource network, to further characterize the mutations associated with RIF-R and investigate the new mutations.

Following the reviewer’s comment, we have revised the manuscript as follow.

1) To) added in Results (2.1. Characteristics of Study Isolates and Profile of rpoB Gene Mutations), P.2

We identified 19 different types of mutations in rpoB gene mutation analysis, with 33 single mutations (33/48, 68.8%) and 15 multiple mutations (15/48, 31.3%). Of the 48 isolates, 17 (17/48, 35.4%) showed a mutation at codon 477 (gct to gat) and presented the amino acid substitution A477D, which was associated with high-level RIF resistance (MIC >128 mg/L).

We found several additional amino acid substitutions of H481Y and S464P that were common mutations of the rpoB proteins and some relatively rare mutations of R484H, S486L, D471G that some previous studies have reported [2,8]. In addition, some isolates in our study carried rare mutations in RRDR, such as Q486L, Q486R, H481R, E568K, R197L.

2) To) added in Discussion, P.6

In the A477D mutant, this substitution places a negatively charged carboxylate unit in close proximity to an existing carboxylate from H481, which is situated near the protein–DNA interface and thereby increases the negative charge on the surface of the protein and destabilizes the enzyme–DNA interaction owing to electrostatic repulsion [39]. Even if the abovementioned substitution does not face the RIF target region, it can induce a conformational change that indirectly prevents antibiotic binding to the target site, which determines the RIF-R [8]. Another interesting finding of our study was that the analyzed isolates lacked the L466S mutation whi h was one of the most commonly found mutations in the clinical isolates of other studies [7,9]. We posit that there should be regional differences in the mutation codon of RIF-R blood S. aureus isolates, given that only limited information is available regarding the molecular characteristics of rpoB mutations of blood S. aureus isolates in multiple countries including South Korea. Future multi-center studies using isolates of national or global pathogen resource networks to further characterize the mutations associated with RIF-R and investigate new mutations is essential.

Comment 2: The authors only sequenced the RIF-R strains so there was no negative control in the study. Perhaps if the authors can sequence a few RIF-S strains to show that they do not have similar rpoB mutations, the study could be greatly strengthened.

Response 2: Thank you for this valuable comment and we totally agree with the reviewer’s concern.

Although we performed molecular typing of 1615 S. aureus isolates that revealed the predominance of ST5 (543/1615, 33.6%) and agr functionality on 1615 S. aureus isolates, the amplification of RRDR of the rpoB gene was performed only for 102 S. aureus isolates (52 RIF-R and 50 RIF-S MRSA strains) in our study. When we sequenced the 50 RIF-S MRSA stains as negative control, we found that there was no detection of amino acid substitutions in any of the 50 RIF-S S. aureus isolates.

However, as the reviewer’s comment, we posit that our study could be strengthened further if we had sequenced more sufficient RIF-R and RIF-S S. aureus isolates, and it has limitation to make a robust confirmation that RIF-R in S. aureus isolates was caused by rpoB mutation.

Following the reviewer’s comment, we tried to lend a more euphemistic tone to the text regarding the association between RIF-R and rpoB mutation, and have revised the manuscript as follow.

1) From) Abstract, P.1

Abstract: Rifampin resistance (RIF-R) in Staphylococcus aureus (S. aureus), possibly from rpoB mutations, has raised concern about clinical-treatment and infection-prevention strategies. Data on the prevalence and molecular epidemiology of RIF-R S. aureus blood isolates in South Korea are scarce.

To) Abstract, P.1

Abstract: Rifampin resistance (RIF-R) in Staphylococcus aureus (S. aureus) with rpoB mutations as one of its resistance mechanisms has raised concern about clinical-treatment and infection-prevention strategies. Data on the prevalence and molecular epidemiology of RIF-R S. aureus blood isolates in South Korea are scarce.

2) From) Results (2.1. Characteristics of Study Isolates and Profile of rpoB Gene Mutations), P.2

Of the 48 isolates, 17 (17/48, 35.4%) showed a mutation at codon 477 (gct to gat) and presented the amino acid substitution A477D, which was associated with high-level RIF resistance (MIC >128 mg/L). However, amino acid substitutions were not detected in any of the 50 RIF-S S. aureus isolates.

To) Results (2.1 Characteristics of Study Isolates and Profile of rpoB Gene Mutations), P.2

Of the 48 isolates, 17 (17/48, 35.4%) showed a mutation at codon 477 (gct to gat) and presented the amino acid substitution A477D, which was associated with high-level RIF resistance (MIC >128 mg/L). However, when we sequenced 50 RIF-S MRSA stains as a negative control, we found that there was no detection of amino acid substitutions in any of the 50 RIF-S S. aureus isolates.

3) To) added in Discussion, P.7

Our study has certain limitations. First, this study was conducted at a single center. Second, we analyzed the clinical outcomes and risk factors of RIF-R S. aureus isolates obtained from patients with bacteremia. Third, we did not evaluate the molecular characteristics of the RIF-R MSSA isolates. We posit that our study could be strengthened further if we had sequenced more sufficient RIF-R and RIF-S S. aureus isolates, and it has limitations in being able to acquire strong confirmation that RIF-R in S. aureus isolates was caused by a rpoB mutation. However, our study included a relatively large number of S. aureus isolates from bacteremia (n = 1615) with RIF-R MRSA isolates (n = 52) and was conducted in a 2700-bed tertiary-care teaching hospital where patients from across the country were admitted during a 10-year period.

4) To) added in Materials and Methods (4.5. PCR Detection of Rifampin Resistance-associated Mutation), P.8

The rifampin resistance-determining region (RRDR) of the rpoB gene sequence, measuring 690 bp (nucleotides 1307–2020), was amplified using PCR. Amplification of rpoB was performed using RRDR1 (5′ TTC AAG ATA CTG AGT CTA TCA CAC C 3′) and rpoB RRDR2 (3′ GCA CG T GAT TCT GGT GCA GCT ATT A 5′) primers, followed by sequence analysis. Amplification was performed for 52 RIF-R and 50 RIF-S strains as previously described [21], and the 50 RIF-S strains were sequenced as a negative control:

Comment 3: From points 1 and 2, in Table 1 there were a few strains with more than 1 mutation. It is rather difficult to identify which mutation (one or more) contributed to the increased MIC.

Response 3: Thank you for this valuable comment. 

As the reviewer’s comment, 15 multiple mutations (15/48, 31.3%) were identified in rpoB gene mutation analysis in our study, all of which were associated with high-level rifampin-resistance (MIC ≥8 mg/L). Our results is consistent with those of several previous studies that revealed the correlation between high-level RIF-R and multiple mutations in rpoB gene [Bahrmand AR et al. J Clin Microbiol 2009;47:2744-2750, Huitric E et al. Antimicrob Agents Chemother 2006;50:2860-2862].

Although most rpoB gene mutations were correlated with high-level resistance [Huitric E et al. Antimicrob Agents Chemother 2006;50:2860-2862, Aubry-Damon H et al. Antimicrob Agents Chemother 1998;42:2590-2594], it is difficult to identify how much each mutation contributed to the increased MIC. We posit that further molecular analysis of rpoB gene mutations in RIF-R S. aureus isolates should be performed to identify the correlation with high-level MIC and mutation at specific codon, which will provide higher robustness on isolates’ ability to survive and affect the clinical practice in the region of endemicity.

We are planning the future multi-center study using isolates of National Culture Collection for pathogens, a national pathogen resource network, to further investigate the significance of multiple mutations in rpoB gene and different levels of RIF-R associated with specific nucleotide replacements.

Following the reviewer’s comment, we have revised the manuscript as follow.

1) To) added in Discussion, P.6

The lack of RRDR mutations in RIF-R MRSA isolates may be due to the presence of other rare rpoB mutations or another mechanism of RIF-R [34], which is supported by previous reports from several countries [35–37]. We posit that the four isolates (7.7%) likely possessed additional mutations outside the RRDR because they exhibited higher RIF-R levels than those conferred by the mutations in their RRDR alleles. On the contrary, 15 multiple mutations (15/48, 31.3%) were identified in rpoB gene mutation analysis in our study, all of which were associated with high-level RIF-R (MIC ≥8 mg/L). Although most rpoB gene mutations were correlated with high-level resistance [6], it is difficult to identify how much each mutation contributed to the increased MIC. Further molecular analysis of rpoB gene mutations in RIF-R S. aureus isolates to identify the correlation between high-level MIC and mutation at specific codons will provide higher robustness on isolates’ ability to survive and affect clinical practice in the region of endemicity.

Comment 4: Line 109 – 110: there are typos (followed by “a” [should be “IVa”], as well as another “a” below).

Response 4: Thank you for this comment. When we described the results of molecular typing, there was mistake that should not be happened. Following the reviewer’s comment, we have revised the typos. 

1) From) Results (2.3. Molecular Typing), P.4

The Staphylococcal chromosomal cassette mec (SCCmec) typing, MLST, and spa typing were performed on the 52 RIF-R MRSA strains (Table 2). SCCmec typing revealed that the most common type was SCCmec type IIb (40/52, 76.9%), followed by a (7/52, 13.5%), â… c (2/52, 3.8%), II (2/52, 3.8%), a (1/52, 1.9%). The results of spa typing revealed diverse distribution of the 52 RIF-R MRSA strains.

To) Results (2.3. Molecular Typing), P.4

The Staphylococcal chromosomal cassette mec (SCCmec) typing, MLST, and spa typing were performed on the 52 RIF-R MRSA strains (Table 2). SCCmec typing revealed that the most common type was SCCmec type IIb (40/52, 76.9%), followed by IVa (7/52, 13.5%), â… c (2/52, 3.8%), II (2/52, 3.8%), and IIa (1/52, 1.9%). The results of spa typing revealed a diverse distribution of the 52 RIF-R MRSA strains.

Comment 5: Some parts of the manuscript are very difficult to understand. For example, Lines 26 - 30 (abstract). Please consult language editors. Also, I noticed the use of the word "pluripotent pathogen", which is rather unusual. Can the authors confirm that this is the right word?

Response 5: Thank you for this valuable comment and we agree with the reviewer’s concern.

Following the reviewer’s comment, we tried once again thoroughly to ensure that our study is written in correct English before submission of revised manuscript.

In addition, as the reviewer’s comment, we have revised the word “pluripotent pathogen” into more appropriate word to express the distinct characteristics of S. aureus as one of the most important pathogens of humans.

1) From) Introduction, P.1

Staphylococcus aureus (S. aureus) is a pluripotent pathogen in both community-acquired and nosocomial infections that causes various illnesses, which range from relatively minor local infections to life-threatening systemic infections [1].

To) Introduction, P.1

Staphylococcus aureus (S. aureus) is one of the most notorious and clinically significant pathogens in both community-acquired and nosocomial infections that causes various illnesses, which range from relatively minor local infections to life-threatening systemic infections [1].

Thank you for your kind and helpful comments.

Round 2

Reviewer 1 Report

Authors has, in a large percentage, accepted the reviewers comments.

Reviewer 4 Report

The authors have sufficiently revised the manuscript according to my comments.

The manuscript now reads better.